# Kerr reversal in Josephson meta-material and traveling wave parametric amplification

Arpit Ranadive [1✉], Martina Esposito [1,2], Luca Planat[1], Edgar Bonet[1], Cécile Naud[1], Olivier Buisson[1], Wiebke Guichard[1] & Nicolas Roch[1]

Josephson meta-materials have recently emerged as very promising platform for superconducting quantum science and technologies. Their distinguishing potential resides in ability to engineer them at sub-wavelength scales, which allows complete control over wave dispersion and nonlinear interaction. In this article we report a versatile Josephson transmission line with strong third order nonlinearity which can be tuned from positive to negative values, and suppressed second order non linearity. As an initial implementation of this multipurpose meta-material, we operate it to demonstrate reversed Kerr phase-matching mechanism in traveling wave parametric amplification. Compared to previous state of the art phase matching approaches, this reversed Kerr phase matching avoids the presence of gaps in transmission, can reduce gain ripples, and allows in situ tunability of the amplification band over an unprecedented wide range. Besides such notable advancements in the amplification performance with direct applications to superconducting quantum computing and generation of broadband squeezing, the in-situ tunability with sign reversal of the nonlinearity in traveling wave structures, with no counterpart in optics to the best of our knowledge, opens exciting experimental possibilities in the general framework of microwave quantum optics, single-photon detection and quantum limited amplification.

[1] Univ. Grenoble Alpes, CNRS, Grenoble INP, Institut Néel, 38000 Grenoble, France. [2] CNR-SPIN, c/o Complesso di Monte S. Angelo, via Cinthia, 80126 Napoli, Italy. ✉email: arpit.ranadive@neel.cnrs.fr

The study of meta-materials[1,2] has generated large interest in the frame of quantum technologies due to wide range of direct applications, e.g., exploration of novel quantum optics phenomena[3–6], non-destructive quantum sensing[7], quantum limited amplification[8–10], or quantum information processing[11–13]. Recently, superconducting Josephson junctions have emerged as very promising building blocks for one dimensional nonlinear meta-materials due to key advantages like low loss, compactness and extraordinarily strong, and tunable nonlinearity[2]. These meta-materials are constructed by embedding an array of Josephson junctions or loops containing Josephson junctions, in a transmission line. The meta-material, fabricated in transmission geometry, can be used as a low noise broadband amplifier out of the box and hence this use case is naturally well suited to demonstrate its efficacy.

Low noise parametric amplifiers underwent an enormous development in the last decade, especially due to their remarkable impact in the field of superconducting quantum information[14]. Resonant parametric amplifiers based on Josephson junctions embedded in a microwave resonator (JPAs)[15,16] provide quantum-limited amplification[17–20] and are widely used for quantum-noise limited microwave readout[21]. The resonant nature of JPAs, however, sets a limit on their bandwidth, and on their applications. Recently, this limitation has been overcome by traveling wave parametric amplifiers (TWPAs) consisting of non-resonant nonlinear transmission lines exhibiting amplification bandwidth up to few GHz[14]. The nonlinearity required for parametric coupling in these devices is provided either by the kinetic inductance of a superconducting material (KTWPAs)[22] or by a Josephson meta-material (JTWPAs)[8–10]. TWPAs presently are very appealing tools for a variety of applications dealing with broadband quantum-noise limited amplification[23], ranging from multiplexed readout of solid state qubits[24–26] to microwave kinetic inductance detectors[27] and dark matter detectors[28,29] for astrophysics, and microwave photonics experiments[12].

Amplification in TWPAs is achieved with a wave mixing process arising from coupling between traveling modes in the nonlinear medium: the medium is excited with a strong pump field, at frequency $\omega_p$, causing the amplification of a weaker signal field, at frequency $\omega_s$, and the creation of an idler field,

at frequency $\omega_i$[30]. Depending on the order of the nonlinearity, the process can be either a three wave mixing process, involving three photons ($\hbar\omega_p = \hbar\omega_s + \hbar\omega_i$), or a four wave mixing process, involving four photons ($2\hbar\omega_p = \hbar\omega_s + \hbar\omega_i$). Both of these processes require specific phase matching conditions, which so far have been realized by employing rather complicated engineering of the dispersion of the TWPA device operating in microwave domain. For three wave mixing TWPAs, dispersion engineering is needed to suppress undesired second order nonlinear processes that would otherwise dominate over the parametric amplification[31]. For four wave mixing TWPAs, dispersion engineering is adopted for maintaining the phase matching condition between pump, signal, and idler across the nonlinear transmission medium, that otherwise would be degraded due to third order phase modulation processes[8–10,22]. Practically, in the TWPA devices demonstrated so far[8,10,22,31], a distortion in the dispersion relation is engineered either by using a photonic-crystal like approach (a photonic gap is opened in the dispersion relation by periodically modulating the transmission line[10,22,31]) or by using a resonant phase matching approach (a stop band gap is created in the dispersion relation by introducing resonant elements in the transmission line[8,9,32]).

The dispersion engineered TWPAs suffer from two main disadvantages. First, the presence of the gap in the dispersion relation typically produces a discontinuous amplification band with significant ripples in the gain profile[8,10,22]. And secondly, the dispersion engineering method only guarantees the optimal amplification conditions when the TWPA is pumped at a designated frequency[8,10,22,31], making the amplification band fixed by design.

In this article, we demonstrate a four wave mixing TWPA with a novel phase matching process based on the sign reversal of the third order (Kerr) nonlinearity and with no need for dispersion engineering[33]. Figure 1 depicts the concept of phase matching with this approach, which overcomes the aforementioned disadvantages of dispersion engineered TWPAs, while maintaining ease of fabrication. Due to absence of dispersion engineering, the device doesn't have gaps in the transmission and a nearly flat impedance across the amplification band. This can significantly reduced the gain ripples (see supplementary information). As the

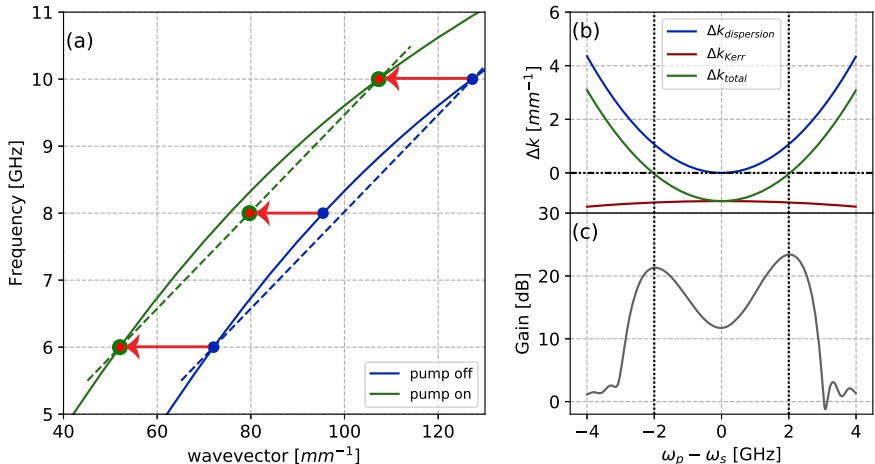

**Fig. 1 Reversed Kerr phase matching. a** Exaggerated dispersion relations of a weak probe tone in presence of pump at 8 GHz (green) and absence of pump (blue) in reversed Kerr TWPA at negative Kerr flux bias. The red arrows indicate the contribution of self phase modulation (SPM, at 8 GHz) and cross-phase modulation (XPM, at 6 and 10 GHz). The XPM experienced by the signal and idler waves is stronger than the SPM experienced by the pump wave, this allows signal, idler and pump to be colinear, i.e., satisfy the phase matching condition. Due to sign reversal of Kerr, self, and cross phase modulation processes cancel phase mismatch arising from curvature in dispersion (blue); which leads to perfect phase matching at 6 and 10 GHz (indicated with red-green points). Simulations of reversed Kerr phase matching and gain (**b**, **c**) using device parameters. The represented $\Delta k$ values are phase mismatches at the output of the devices.

sign reversal of the Kerr nonlinearity is not frequency dependent, the phase-matched amplification band can be dynamically tuned in situ by simply changing the pump frequency.

The meta-material, operated as reversed Kerr TWPA, exhibits up to 4 GHz combined bandwidth, 8 GHz tunability of the amplification band, −98 dBm saturation (1 dB compression) at 20 dB gain and added noise near the standard quantum limit.

Our results prove that nonlinear meta-materials constructed from Josephson junctions, in contrast to their optical counterparts (nonlinear optical fibers[34–36]), can be tailored to achieve in-situ tunability of the Kerr nonlinearity; opening the door for the exploration of this nonlinear optics regime, e.g., generation of broadband non-classical states[12] and realization of non destructive traveling photon counters[7].

## Results

**Device description and operation.** The device presented in this article is constituted of 700 cells spanning 6 mm, with each cell containing a superconducting loop with three large (high critical current, $I_0$) and one small (low critical current, $rI_0$) Josephson junctions in either arm. This design is known in the super-conducting circuit community with the name of superconducting nonlinear asymmetric inductive element (SNAIL)[37], and has been successfully adopted for three wave mixing resonant parametric amplification[38–40] and the implementation of Kerr-cat qubits[41]. The use of this circuit element, and in general of elements giving rise to both second and third order nonlinearities, for traveling wave parametric amplification has been proposed/investigated in the last five years with different approaches[33,42–47]; the device presented here is, to our knowledge, the first experimental demonstration of a reversed Kerr traveling wave parametric amplifier.

The asymmetry between the two arms of the superconducting loop allows to have both even and odd nonlinear terms in the Taylor expansion of the current-phase relation:

$$I(\phi) \approx \frac{\Phi_0}{2\pi L}\phi - 3\Phi_0\sqrt{\frac{R_Q}{\pi^3 Z^3}}g_3\phi^2 - 4\Phi_0\frac{R_Q}{\pi^2 Z^2}g_4\phi^3, \quad (1)$$

where $g_3$ and $g_4$ are the three wave and four wave mixing flux-tunable nonlinear coefficients indicating rates at which corresponding interaction manifests, $\Phi_0$ the magnetic flux quantum, $R_Q = h/4e^2$, $Z$ (defined as $\sqrt{L/C_g}$) is the characteristic impedance of the transmission line, with $L$ the flux-tunable inductance per unit cell and $C_g$ the ground capacitance per unit cell. Explicit expressions for $L$, coefficients $g_3$ and $g_4$ as a function of the external magnetic flux are given in the methods.

The ratio of the critical current of small and large Josephson junction ($r$) is chosen such that the flux dependent magnitudes of the second order nonlinearity ($g_3$) and third order nonlinearity ($g_4$) are anti-correlated. In addition, the device is designed to ensure that the third order nonlinearity ($g_4$) attains sufficiently large negative values. The anti-correlated nature of $|g_3|$ and $|g_4|$, meaning that when one is zero the other is maximum, allows the separate exploration of the desired wave mixing processes. To further suppress spurious three wave mixing processes, we use SNAIL elements with alternating magnetic flux polarity, i.e., they are physically oriented in a way such that magnetic flux biasing is reversed for adjacent SNAIL cells, as depicted in Fig. 2. Since $g_3$ is an odd function of the external flux, its value in adjacent cells has opposite sign resulting in an overall cancellation of three wave mixing processes at the wavelength scales under discussion in this article[48]. To illustrate this suppression, a comparison between simulated second harmonic generation in the meta-material with and without the polarity inversion is shown in supplementary information. The nonlinear coefficient $g_4$ is shown in Fig. 2a as a

function of the external magnetic flux. In this work, we focus on using the device as four wave mixing reversed Kerr parametric amplifier, and for this purpose we flux-tune the device in order to have maximally negative $g_4$.

The coupled evolution of signal and idler fields in presence of a strong pump in the nonlinear medium tuned to exhibit maximally negative $g_4$ can be described by the following wave equations[8,49],

$$\left(\partial_x + \frac{i\Delta k}{2}\right)a_s = i\frac{k_i}{2k_s}\eta_s a_i^\dagger - \kappa_s'' a_s, \quad (2)$$

$$\left(\partial_x - \frac{i\Delta k}{2}\right)a_i^\dagger = -i\frac{k_s}{2k_i}\eta_i a_s - \kappa_i'' a_i^\dagger, \quad (3)$$

where $\eta_{s,i}$ are $g_4$ dependent coupling constants, $k_{s,i}$ are the real component of signal and idler wavevectors, $\Delta k$ is the total phase mismatch, and $\kappa_{s,i}''$ are the imaginary component of signal and idler wavevectors. With the approximation of lossless transmission line ($\kappa_{s,i}'' = 0$) and treating $a_{s,i}$ as semi-classical fields with zero initial idler boundary condition, we get the familiar expression[8,10,32,50] for the non-degenerate four wave mixing power gain,

$$G_{\text{power}} = \cosh^2(gx) + \frac{\Delta k^2}{4g^2}\sinh^2(gx), \quad (4)$$

where $g$ is the reduced gain coefficient defined in[33] and $x$ is position along the TWPA. The gain that can be achieved with such coupled evolution is maximum when $\Delta k$ vanishes. The total phase mismatch can be expressed as the sum of two contributions:

$$\Delta k = \Delta k_{\text{dispersion}} + \Delta k_{\text{Kerr}}. \quad (5)$$

The linear phase mismatch between signal, idler and pump fields, $\Delta k_{\text{dispersion}} = k_s + k_i - 2k_p$, is defined by the dispersion relation

$$k(\omega) = \frac{\omega}{\omega_0\sqrt{1 - \omega^2/\omega_J^2}}, \quad (6)$$

which is obtained by solving for transmission in absence of second and third order nonlinearities ($g_3 = g_4 = 0$),

$$\begin{aligned} &\frac{\partial^2\phi}{\partial x^2} - \frac{1}{\omega_0^2}\frac{\partial^2\phi}{\partial t^2} + \frac{1}{\omega_J^2}\frac{\partial^4\phi}{\partial x^2\partial t^2} \\ &+ 6g_3\sqrt{\frac{R_Q}{\pi\omega_0^2 Z}}\frac{\partial}{\partial x}\left[\left(\frac{\partial\phi}{\partial x}\right)^2\right] \\ &- 8g_4\frac{R_Q}{\pi\omega_0 Z}\frac{\partial}{\partial x}\left[\left(\frac{\partial\phi}{\partial x}\right)^3\right] = 0, \end{aligned} \quad (7)$$

where $\omega_J = \sqrt{1/LC_J}$, $\omega_0 = \sqrt{1/LC_g}$, denote the plasma frequency and the characteristic frequency of the transmission line, respectively. It can be noted that Eq. (7) is semi-classical approximation of Eqs. (2) and (3). The non linear phase mismatch, $\Delta k_{\text{Kerr}}$, describes the phase mismatch arising due to the third order phase modulation processes (self-phase modulation and cross-phase modulation) and linearly depends on the pump power and the nonlinear coefficient $g_4$. Following the approach used in ref. [33], one can show that the sign of $\Delta k_{\text{Kerr}}$ only depends on the sign of $g_4$. See Eqs. (21) and (22) in methods for expressions of $\Delta k_{\text{Kerr}}$.

The four wave mixing TWPAs demonstrated so far work in the regime of positive third order (Kerr) nonlinearity ($g_4 > 0$). In that case, the phase matching is accomplished by introducing a gap in the transmission line and changing the sign of $\Delta k_{\text{dispersion}}$ for a specific pump frequency corresponding to the gap position. In the

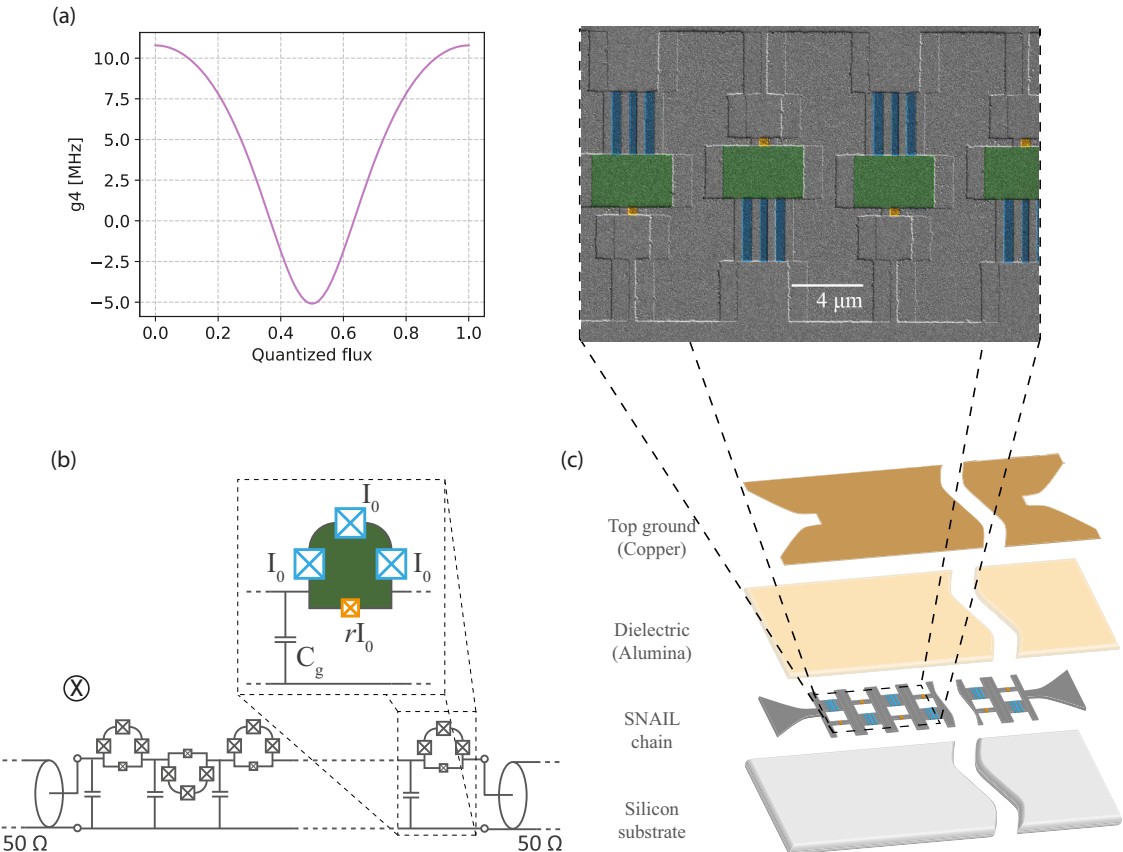

**Fig. 2 Reversed Kerr TWPA implementation. a** Nonlinear coefficient $g_4$ as a function of the external magnetic flux, calculated using the device parameters obtained from the linear characterization ($I_0 = 2.19\,\mu$A and $r = 0.07$, see supplementary information). **b** Circuit schematic of the reversed Kerr TWPA; $I_0$ is the critical current of the large junction, $rI_0$ is the critical current of the small junction and $C_g$ is the ground capacitance per unit cell. The device is threaded by a global magnetic field. The alternated loop geometry hence results in an alternated magnetic flux polarity. **c** Schematic of the reversed Kerr TWPA. The device is fabricated with double angle aluminum Josephson junction array deposition on silicon substrate, followed by atomic layer deposition of dielectric (alumina) and finally by gold top ground deposition[51]. A scanning electron microscopy (SEM) image of the meta-material with false coloring to indicate large (blue) and small (orange) junctions forming a SNAIL in the transmission line is shown as a closeup; the superconducting loop area is colored green.

Reversed Kerr TWPA presented here, the need for dispersion engineering is circumvented by operating the device in the regime of negative third order (Kerr) nonlinearity ($g_4 < 0$), i.e., negative $\Delta k_{\text{Kerr}}$.

Exaggerated dispersion relations of a weak probe tone in presence of pump at 8 GHz (green) and absence of pump (blue) in reversed Kerr TWPA at negative Kerr flux bias are shown in Fig. 1a. Panel b shows simulated $\Delta k_{\text{dispersion}}$ (blue), $\Delta k_{\text{Kerr}}$ (red) and the total phase mismatch (green, sum of the two) as a function of separation of signal frequency from pump frequency; using the device parameters. Due to reversed sign of the Kerr phase modulation, total phase mismatch vanishes at two signal frequencies, symmetrically on either side of the pump frequency; panel (c) depicts simulated gain for such phase mismatch; one can notice that the gain maxima correspond to the frequencies with optimal phase matching. The positions of these gain maxima are dependent on the device design parameters, and can be changed by modifying the nonlinear coefficients and dispersion relation.

The fabrication steps of the reversed Kerr TWPA are depicted in Fig. 2c. The device chain containing Josephson junctions is fabricated on intrinsic silicon substrate using double angle evaporation method with aluminum, followed by atomic layer deposition (ALD) of alumina dielectric and gold top ground[51]. The thin (30 nm) dielectric layer of Alumina provides the ground capacitance required for matching impedance of the transmission line to standard 50 Ω environment[51]. This junction fabrication method is expected to yield junction disorder(standard deviation)

lower than 3%[52], which is within tolerable limit for realizing a high performance reversed Kerr TWPA[33].

3D finite element simulations of the magnetic field generated by the current carrying coil with 3 cm diameter, mounted on top of the sample holder, and used to bias the TWPA device, predict field asymmetry lower than 0.1% across the chain of SNAIL loops.

**Reversed Kerr gain characterization.** The device is experimentally characterized in a dilution refrigerator at 20 mK with standard microwave electronics (See supplementary information). We focus on the amplification performance of the device operated at half flux (in the reversed Kerr regime). The gain characteristics for the device are depicted in Fig. 3, measured as the difference in transmission when pump is on and when pump is off. Panel a shows the gain obtained by pumping the same device at different frequencies. When the device is pumped at 6 GHz, it exhibits a gain around 15 dB over 5 GHz combined bandwidth, pumped at 8 GHz it exhibits a gain around 18 dB over 3.5 GHz combined bandwidth, and when pumped at 10 GHz it exhibits a gain larger than 20 dB over 4.5 GHz combined bandwidth. It should be noted that we quote the combined bandwidth of the two amplification bands on either side of the pump, as this is the most relevant number for most of the intended applications of the device. Josephson TWPAs are very efficient at amplification and need low pump powers to operate. Similar pump power

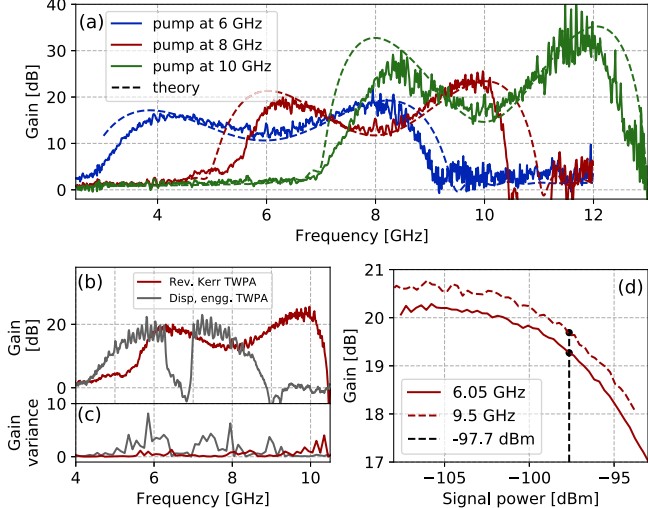

**Fig. 3 Reversed Kerr gain. a** Gain profile of same device pumped at 6, 8, and 10 GHz and tuned to operate at external magnetic flux $\Phi_{ext}/\Phi_0 = 0.5$. Dashed lines indicate the gain predicted by the theoretical model. **b, c** Comparison of gain and gain variance of reversed Kerr TWPA pumped at 8 GHz to TWPA with phase matching implemented using stop band engineering[10] pumped to 6.635 GHz. The variance has been evaluated over bins of 100 MHz. **d** Gain compression as a function of input signal power when the reversed Kerr TWPA was pumped at 8 GHz, at signal frequencies on both side of pump with gain close to 20 dB. The 1 dB compression point at −97.7 dBm is indicated with a dashed line.

($\sim$ −75 dBm) was used at the input of the TWPA to obtain optimal gain profiles at various frequencies depicted in Fig. 3. The double lobe gain curves result from the fact that phase mismatch diminishes on either side of the pump as depicted in Fig. 1. The curvature of the dispersion relation leads to higher gain at higher frequencies without sacrificing bandwidth. This is direct consequence of strong dependence of the reduced gain coefficient ($g$ in Eq. (4)) on pump, signal and idler wavevectors[10,33]; $g$ increases with increase in wavevectors. Since wavevectors of traveling modes in the TWPA are proportional to their frequencies and diverge at the plasma frequency, higher gain is expected and observed at higher frequencies. This can also be understood as increase in effective interaction time of the modes in the TWPA due to the reduction in phase velocity.

The theoretical plots corresponding to the experimental gain curves are also presented (dashed lines); they are obtained by solving coupled differential Eqs. (2) and (3), and summarized in the methods. The magnitude of gain is in good qualitative agreement with the theoretical model. We stress that the model has no fitting parameters but just utilizes the values of $C_J$ (Josephson capacitance per unit cell) and $C_g$ (ground capacitance per unit cell) obtained by design, the values of $r$ and $I_0$ obtained from the device characterization in linear regime (details in supplementary information) and the experimental values for the pump power. It can be noted that the gain obtained at higher frequencies for a fixed pump frequency is consistantly higher. The gain obtained at pairs of frequencies equidistant from the pump, in an ideal lossless reversed Kerr TWPA would be identical, however, practically, asymmetric loss experienced by waves traveling at different frequencies breaks this gain symmetry. This dynamics is well captured by the distributed gain model used to simulate theoretical plots. One possible explanation for the slight narrowing of the amplification band with respect to the model prediction could be stray linear inductance in the transmission line, which is not accounted in the model. Overall, we observed a

combined dynamical bandwidth larger than 10 GHz with gain higher that 15 dB, which is one of the distinguishing features of this device.

Another significant merit of the presented meta-material operated in reversed Kerr regime for amplification, is the absence of gaps and dispersion engineering in the transmission band, which reduces gain ripples significantly. A comparison of characteristic impedance of transmission lines with and without dispersion engineering, and how it translates to gain ripples is discussed in supplementary information. Panel c in Fig. 3 compares the gain profile of the reversed Kerr TWPA pumped at 8 GHz with a band engineered TWPA with a similar fabrication technique and pumped at 7 GHz[10]. Panel d in Fig. 3 depicts the contrast in gain variance arising from the aforementioned ripples. Identical packaging (sample holders and connectors) was used for both devices and the gain variance was evaluated over bins of 100 MHz. Lower gain ripples also contribute in improving gain stability.

The operating point of reversed Kerr TWPA corresponds to an extremum of third order nonlinearity with respect to magnetic flux threaded through the SNAIL loop, which makes it ($g_4$) and phase matching condition first order insensitive to magnetic flux variations offering a natural protection. We observed time domain gain variance between 0.5–1.5 dB over 18 h of continuous measurement using standard commercial microwave generators and without magnetic shielding for the device. With moderate magnetic shielding, the stability can be further improved.

The saturation power is also investigated by measuring the gain versus the input signal power. Panel c in Fig. 3 depicts gain saturation curve for the device when pumped at 8 GHz. We observed 1 dB gain compression (saturation) at −97.7 ± 0.5 dBm for gain around 20 dB. This is comparable to state of the art in JTWPAs[8,10]. Input and output line calibration used for obtaining the same is described in the methods.

**Noise calibration**. The noise calibration for the TWPA has been performed using a broadband thermal noise source with temperature tunability between 40 mK to 1 K. Using this noise source, we first calibrated the noise and gain performance of the output line (without TWPA). The output noise power of the transmission line including TWPA, in the high TWPA gain limit can be expressed as,

$$P(\omega_s) = \big\{ N_{source}(\omega_s, T_{source})\, G_{TWPA}(\omega_s) + N_{TWPA}(\omega_s)\, G_{TWPA}(\omega_s)$$
$$+ N_{source}(\omega_i, T_{source})\, G_{TWPA}(\omega_i) + N_{out}(\omega_s) \big\}\, G_{out}(\omega_s) B_w,$$
$$\tag{8}$$

where $N_{source}(\omega, T_{source})$ is the noise emitted by the thermal noise source at frequency $\omega$ when heated to temperature $T_{source}$, $G_{TWPA}(\omega)$ and $N_{TWPA}(\omega)$ correspond to the TWPA gain and added noise, $G_{out}(\omega)$ and $N_{out}(\omega)$ correspond to output line (without TWPA) gain and added noise, at frequency $\omega$; $B_w$ denotes measurement bandwidth. It should be noted that expression (8) includes input noise at both signal and idler frequencies, since we are using a broadband noise source.

The noise temperature of the TWPA pumped at 8 GHz, obtained by fitting the measured power spectral density to the signal-idler two mode model in Eq. (8), is shown in panel (a) of Fig. 4 along with system noise temperature and standard quantum limit for non-degenerate amplification. Due to broadband nature of the noise source and high bandwidth of the amplifier, the power emitted by noise sources reaches close to saturation power of amplifier for temperatures above 0.5 K. Thus, we limit the temperature range of the noise fit to 400 mK to avoid any artifacts of amplifier saturation. At peak TWPA gains, we observed the noise added by the TWPA to be around twice the

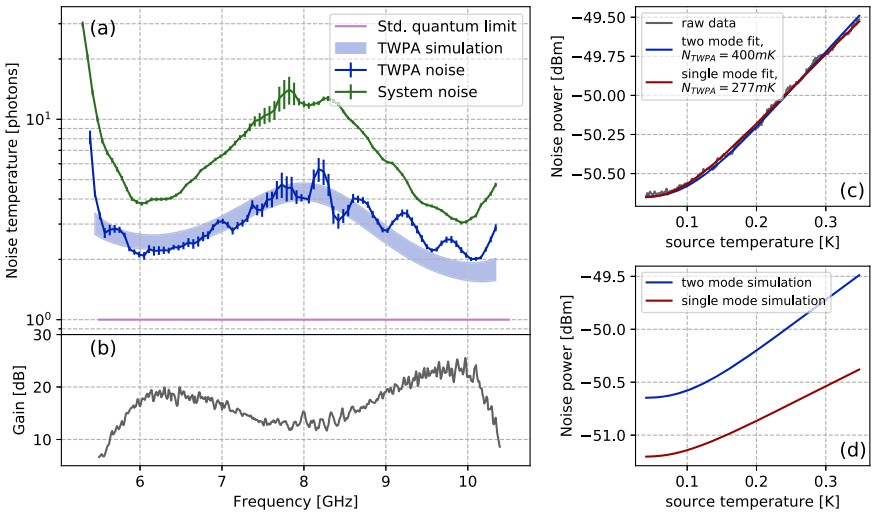

**Fig. 4 Noise characterization. a** Noise temperature of the amplification chain (green) and the reversed Kerr TWPA (blue) compared to the standard quantum limit (SQL) across amplification band in terms of number of added photons. The device is pumped at 8 GHz and tuned to external magnetic flux $\Phi_{\text{ext}}/\Phi_0 = 0.5$. The blue shadow area indicates the result of the theoretical simulations considering power dependent losses. **b** Corresponding gain profile. **c** Comparison between output noise power fitted to noise models including and excluding idler mode noise input. Ignoring idler mode inaccurately leads to lower noise temperature estimates. **d** Comparison of simulated noise output for the noise models including and excluding idler mode with identical TWPA noise temperature.

standard quantum limit. We also plot the theoretical simulation of the noise obtained by numerically solving the coupled differential Eqs. (2) and (3), along with noise source term[49] arising from dielectric losses. As these losses are power dependent, we plot the theoretical simulation results as the band between low power and high power losses (see "Methods" section). We note that the theoretical simulation is in good agreement with the measured noise, indicating that the deviation from the standard quantum limit is primarily due to the dielectric losses in the TWPA device. A promising avenue for improvement of noise performance can be substituting alumina with a low loss dielectric (for example SiN$_x$ or a:SiH[53]), however, that is beyond the scope of this article.

We emphasize that it is crucial to include both signal and idler modes in the noise model whenever a broadband noise source is used for noise calibration, otherwise the added noise could be misinterpreted to be better than the actual value by up to a factor of two[31,54]. Panels c and d of Fig. 4 explain in details this aspect of the noise characterization. Panel c depicts a comparison of noise temperature data at 6.4 GHz fitted to the simple single mode model (giving $N_{\text{TWPA}} = 277$ mK) and signal-idler two mode model (giving $N_{\text{TWPA}} = 400$ mK). Although both the models seem to fit the data, the simple single mode model would give an underestimated noise temperature. In addition, panel d depicts a comparison of simulated system output noise at 6.4 GHz with the simple single mode model and signal-idler two mode model. When only the signal mode is considered, the gain is misinterpreted to be higher than the actual value, which leads to inaccurate normalization of the added noise.

## Discussion

We have presented a Josephson meta-material embedded transmission line with in-situ sign reversal of Kerr nonlinearity. To demonstrate its efficacy, we have shown its use case as a TWPA with a fundamentally new phase matching mechanism: reversed Kerr phase matching.

Conversely to previously demonstrated TWPAs, the absence of gaps in transmission can provide continuous amplification band with significantly lower gain ripples. Also, it provides in-situ

tunability of the amplification band over an unprecedented range (gain larger than 15 dB is observed in the entire 4–12 GHz range) by simply changing the pump frequency. The latter constitute a notable advantage with respect to previous state of the art TWPAs, where the pump frequency is constrained by the dispersion engineering approach. By exploiting a new nonlinear optics phase matching mechanism (reversed Kerr phase matching), presented device is extremely promising for a wide range of amplification applications including multiplexed qubit readout, KID arrays and dark matter detection.

Apart from amplification, Josephson junction based nonlinear devices have also emerged as a very promising platform for novel experiments like generation of non-classical states and photon detectors, so far demonstrated with resonant structures[54–58]. Traveling wave structures with in situ control of the second and the third order nonlinearities pave the way for broadband implementations like multi-mode squeezing experiments and non-destructive traveling wave photon counters[7,12,59].

## Methods

**Device unit cell.** The unit cell of the device is composed by a superconducting loop with three large junctions in one arm and one small junction in the other arm, and a shunt capacitance to ground $C_g$. Indicating the phase difference across the small junction as $\phi$ and using the flux quantization for a single loop we get that

$$\phi_L = \frac{\phi - \phi_{\text{ext}}}{3}, \tag{9}$$

where $\phi_L$ is the phase difference across the large junction and $\phi_{\text{ext}} = 2\pi\Phi_{\text{ext}}/\Phi_0$ is the reduced external magnetic flux. Thus, the current through the asymmetric SQUID can be written as

$$I(\phi) = rI_0 \sin\phi + I_0 \sin\left(\frac{\phi - \phi_{\text{ext}}}{3}\right), \tag{10}$$

where $I_0$ is the large junction critical current and $r < 1$ is the asymmetry ratio.

We can then perform a Taylor expansion about a flux $\phi^*$ such that $I(\phi^*) = 0$ (steady state),

$$\frac{I(\phi^* + \phi)}{I_0} = \frac{dI}{d\phi}\Big|_{\phi^*}\phi + \frac{1}{2}\frac{d^2I}{d\phi^2}\Big|_{\phi^*}\phi^2 + \frac{1}{6}\frac{d^3I}{d\phi^3}\Big|_{\phi^*}\phi^3 + \dots \tag{11}$$

We obtain the following approximated expression,

$$\frac{I(\phi^* + \phi)}{I_0} \approx \widetilde{\alpha}\,\phi - \widetilde{\beta}(\phi)^2 - \widetilde{\gamma}(\phi)^3, \tag{12}$$

where

$$\widetilde{\alpha} = r\cos\phi^* + \frac{1}{3}\cos\left(\frac{\phi^* - \phi_{\text{ext}}}{3}\right), \tag{13}$$

$$\widetilde{\beta} = \frac{1}{2}\left[r\sin\phi^* + \frac{1}{9}\sin\left(\frac{\phi^* - \varphi_{\text{ext}}}{3}\right)\right], \tag{14}$$

$$\widetilde{\gamma} = \frac{1}{6}\left[r\cos\phi^* + \frac{1}{27}\sin\left(\frac{\phi^* - \phi_{\text{ext}}}{3}\right)\right]. \tag{15}$$

The inductance per unit cell can be approximated by keeping terms up to first order in Eq. (11),

$$L = \frac{\Phi_0}{2\pi I_0 \widetilde{\alpha}}. \tag{16}$$

The nonlinear coefficients $g_3$ and $g_4$ in the main text are defined as

$$\hbar g_3 = \frac{\widetilde{\beta}}{3\widetilde{\alpha}}\sqrt{E_C \hbar \omega_0}, \qquad \hbar g_4 = \frac{\widetilde{\gamma}}{2\widetilde{\alpha}}E_C, \tag{17}$$

and where $\omega_0$ is the characteristic frequency of the transmission line defined as $1/\sqrt{LC_g}$ and $E_C$ is the charging energy defined as $e^2/2C_g$.

The circuit parameters for the unit cell are: $C_g = 250$ fF, $C_J = 50$ fF, $I_0 = 2.19$ μA and $r = 0.07$; see supplementary information for linear characterization of the device.

**Transmission loss in low power regime**. Transmission loss of the device for two selected flux values, $\Phi_{\text{ext}}/\Phi_0 = 0$ and $\Phi_{\text{ext}}/\Phi_0 = 0.5$ is depicted in Fig. 5 in low power (~1 photon) regime.

See supplementary information for a full characterization of the device in the linear regime.

**Four wave mixing amplification**. Focusing on the half-flux point, with $g_3 = 0$, three wave mixing non linear processes can be neglected. We numerically solve coupled differential Eqs. 2 and 3 of main text to simulate gain. The equations can be expresses in matrix form as[8,49],

$$\hat{A}'(x) = \Phi(x)\hat{A}(x), \tag{18}$$

where $\hat{A} = [a_s, a_i^\dagger]^T$ and the coordinate $x$ is normalized to number of SNAILs. The matrix $\Phi(x)$ is defined as follows,

$$\Phi(x) = \begin{bmatrix} -i\frac{\Delta k}{2} - \kappa_s'' & i\frac{k_i}{2k_s}\eta_s \\ -i\frac{k_s}{2k_i}\eta_i & i\frac{\Delta k}{2} - \kappa_i'' \end{bmatrix}, \tag{19}$$

where $\kappa_s''$ and $\kappa_i''$ are imaginary components of wavevector at signal and idler frequencies, respectively, and,

$$\Delta k = \Delta k_{\text{dispersion}} + \Delta k_{\text{Kerr}}, \tag{20}$$

$$\Delta k_{\text{Kerr}} = \eta_s + \eta_i - 2\eta_p, \tag{21}$$

$$\eta_{s,i} = \frac{6\gamma}{8\bar{\omega}_{s,i}}k_p^2 k_{s,i}|A_p|^2, \qquad \eta_p = \frac{3\gamma}{8\bar{\omega}_p}k_p^3|A_p|^2, \tag{22}$$

where $|A_p|$ is the pump amplitude and $\widetilde{\omega}_m = \left(1 - \omega_m^2/\omega_J^2\right)$. We approximate the

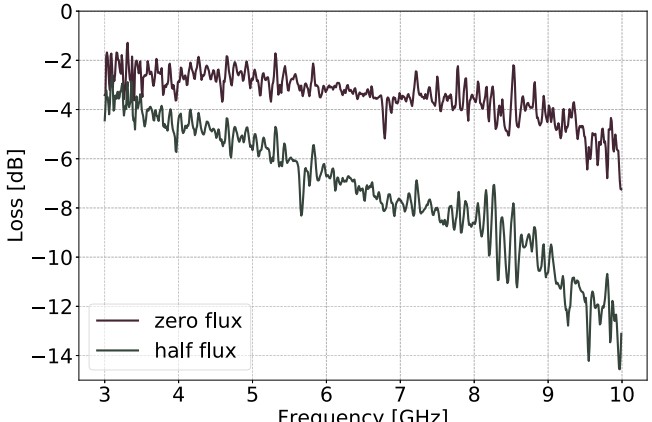

**Fig. 5 Transmission loss.** Transmission loss of the device for two selected flux values, $\Phi_{\text{ext}}/\Phi_0 = 0$ and $\Phi_{\text{ext}}/\Phi_0 = 0.5$ in low power (~1 photon) regime.

effect of pump attenuation as a position independent amplitude reduction, resulting in reduced coupling constant, $|A_p| = A_{p0}\exp(-\kappa_p''N/2)$[8]. The solution of the matrix differential equation, Eq. (18), can be expressed as,

$$\hat{A}(x) = S(x)\hat{A}(0), \qquad S(x) = e^{\Phi x}. \tag{23}$$

We use the $S(x)$ matrix to numerically compute the theoretical gain depicted in Fig. 3 of the main text. It is necessary to include the transmission losses of the device into the gain modeling for accurate quantitative simulation of the gain. Figure 4 in supplementary information shows comparison between simulated gain obtained from lossless approximation and complete model accounting for dielectric losses.

**Setup calibration**. A sketch of the experimental setup is shown in Fig. 1 in supplementary information. We use two cryogenic microwave switches in situ switch between the TWPA sample and a reference box (with TWPA chip replaced by a PCB CPW transmission line). This allows the measurement of phase and power references for input–output lines in a single cool-down cycle.

For noise calibration, we use a thermal noise source, coupled to the setup with a directional coupler (Fig. 1 in supplementary information). The noise source consists in an impedance matched resistive termination mounted to a copper mount with controllable temperature and connected to the directional coupler with a superconducting coaxial cable[60]. The Johnson noise spectra of the noise source is controlled by changing the temperature of the resistive termination.

To calibrate the output line (excluding TWPA), we move the microwave switch to reference position and record the power spectral density (PSD) with a spectrum analyzer as a function of the temperature of noise source, over the desired frequency band. We fit the obtained PSD to standard noise model,

$$P(\omega_s) = \{N_{\text{source}}(\omega, T_{\text{source}}) + N_{\text{out}}(\omega)\}\, G_{\text{out}}(\omega)B_w, \tag{24}$$

where, $G_{\text{out}}(\omega)$ and $N_{\text{out}}(\omega)$ are the fit parameters corresponding to the output line (excluding TWPA) gain and added noise, at frequency $\omega$, $B_w$ denotes measurement bandwidth, and $N_{\text{source}}(\omega, T_{\text{source}})$ is the noise emitted by the thermal noise source at frequency $\omega$, when heated to temperature $T_{\text{source}}$,

$$N_{\text{source}}(\omega, T_{\text{source}}) = \frac{\hbar\omega}{2}\coth\left(\frac{\hbar\omega}{2k_B T_{\text{source}}}\right). \tag{25}$$

We use this output line calibration in the two mode TWPA noise model, as described in the main text, to perform TWPA noise characterization.

The output line gain is also used to obtain the input line attenuation, by subtracting it from the round trip power transmission of the reference box.

**Noise model**. In this session we give details on the derivation of the noise model (Eq. (7) in the main text) used to get the TWPA added noise ($N_{\text{TWPA}}(\omega)$) from the measured power spectral density and using the calibrated noise source as input noise.

When a broadband noise source is used for noise calibration, the amplifier is subjected to incident photons at both signal and idler frequencies. Four wave mixing parametric amplifiers working in phase preserving (non-degenerate) regime couple signal and idler modes. The noise incident at idler mode can result in generation of significant photons at signal frequency and vice versa; this needs to be carefully accounted for in the noise model. An incident field $a_{\text{in,S}}$ at signal frequency, generates an amplified response $\sqrt{\widetilde{G}_S}\, a_{\text{in,S}}$ at signal frequency and a secondary response $\sqrt{\widetilde{G}_S - 1}\, a_{\text{in,S}}^\dagger$ at idler frequency, where $\widetilde{G}_S$ is the gain at signal frequency. The outgoing field at signal frequency from the TWPA can be expressed with the following scattering relation,

$$a_{\text{out,S}} = \sqrt{\widetilde{G}_S}\, a_{\text{in,S}} + \sqrt{\widetilde{G}_I - 1}\, a_{\text{in,I}}^\dagger, \tag{26}$$

where $\widetilde{G}_i$ is the gain at idler frequency.

In order to take into account losses, we can model the TWPA as a combination of an amplifier with gain $\widetilde{G}$ and an attenuator with attenuation $\kappa$. In the beam-splitter loss model[49] $\kappa$ indicates the reflection coefficient of the beam splitter. In high gain limit, we get,

$$\begin{aligned} a_{\text{out,S}} = \; & \sqrt{\widetilde{G}_S\,\kappa_S}\, a_{\text{in,S}} + \sqrt{\widetilde{G}_S(1-\kappa_S)}\, a_{\text{attn,S}}, \\ & + \sqrt{\widetilde{G}_I\,\kappa_I}\, a_{\text{in,I}}^\dagger + \sqrt{\widetilde{G}_I(1-\kappa_I)}\, a_{\text{attn,I}}^\dagger \end{aligned} \tag{27}$$

where $a_{\text{attn}}$ is the field originated from the attenuation (second input field in the beam-splitter model). Using the output field from this expression to calculate the power spectral density and modeling $a_{\text{attn}}$ field as the source of the added noise in the TWPA, we arrive at Eq. (7) of the main text.

**Noise simulation**. In this session we give the details of the simulation of the TWPA noise temperature (blue shaded area in Fig. 4 of the main text) obtained considering the device dielectric losses as main source of the TWPA added noise.

One of the primary sources of extra noise (above standard quantum limit) added by the the TWPA is dielectric losses in the nonlinear transmission line[51].

These losses originate from the presence of a thin alumina layer, necessary to engineer the high capacitance needed for maintaining impedance matching to the environment across the device. We obtain the transmission loss in the device by comparing the transmission of the device with that of the reference box. Inset in Fig. 2 in Supplementary Information depicts the frequency dependence of the loss at operation (half flux) point, which is used to obtain the the imaginary component of the wavevector ($\kappa''$).

To simulate the noise added by the TWPA, we numerically solve coupled differential equations for signal and idler modes with source terms accounting for losses[49],

$$\left(\partial_x + \frac{i\Delta k}{2}\right)a_s = i\frac{k_i}{2k_i}\eta_s a_i^\dagger - \kappa_s'' a_s + \sqrt{\kappa_s''}a_s^{(\mathrm{loss})}, \tag{28}$$

$$\left(\partial_x - \frac{i\Delta k}{2}\right)a_i^\dagger = -i\frac{k_s}{2k_i}\eta_i a_s - \kappa_i'' a_i^\dagger + \sqrt{\kappa_i''}a_i^{\dagger(\mathrm{loss})}. \tag{29}$$

The discretized solution for a transmission line with $N$ SNAILs can be expressed as,

$$\hat{A}(x) = S(x)\hat{A}(0) + \frac{1}{2\pi}\sum_x^N S(N-x)\begin{bmatrix}\sqrt{\kappa_s''}a_s^{(\mathrm{loss})} \\ \sqrt{\kappa_i''}a_i^{\dagger(\mathrm{loss})}\end{bmatrix}, \tag{30}$$

where $S(x)$ is the solution matrix obtained in Eq. (23), without source terms.

Using above solution, and assuming good line thermalization ($\langle\{a_s^{(\mathrm{loss})}, a_s^{\dagger(\mathrm{loss})}\}\rangle = 1/2$), we estimate the noise added by TWPA by calculating $\langle\{a_s^\dagger, a_s\}\rangle$.

Since the transmission losses in the device are power dependent, in Fig. 4a of main text we plot a range between noise estimates calculated using low power losses (~1 photon) and high power losses (~100 photons). The approximate photon numbers are calculated as product of photon flux with travel time in TWPA.

## Data availability

The data supporting the results presented in this article are available at Zenodo open-access repository under https://doi.org/10.5281/zenodo.5647785.

## Code availability

The python scripts used for simulation of gain and dispersion through meta-material are available at github repository JJ metamaterial simulation.

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

## Acknowledgements

This project has received funding from the European Union's Horizon 2020 research and innovation programme under grant agreement no. 899561 and by the ANR under contract BOCA (project number ANR-18-CE47-0003). A.R. acknowledges the European Union's Horizon 2020 research and innovation programme under the Marie Sklodowska-Curie grant agreement No 754303 and the 'Investissements d'avenir' (ANR-15-IDEX-02) programs of the French National Research Agency. M.E. acknowledges the European Union's Horizon 2020 research and innovation programme under the Marie Sklodowska-Curie (grant agreement no. MSCA-IF-835791). The samples were fabricated in the clean room facility of Institute Neel, Grenoble, we thank the clean room staff and L. Cagnon for help with fabrication of the devices. We would like to acknowledge E. Eyraud for his extensive help in the installation of the experimental setup. We thank J. Jarreau, L. Del Rey, D. Dufeu, F. Balestro, and W. Wernsdorfer for their support with the experimental equipment. We are also grateful to F. Lecocq, J. Aumentado, B. Boulanger, and members of the superconducting circuits group at Neel Institute for helpful discussions. We sincerely thank R. Vijayaraghavan and M. Devoret for their careful reading of the manuscript.

## Author contributions

A.R., L.P., and N.R. conceptualized the experiment, A.R. fabricated and measured the devices with the help of M.E. and analyzed the data with the help of M.E., L.P., and N.R., while E.B., C.N., O.B., and W.G. provided support in setting up the experimental platforms and in the interpretation of the experimental results. A.R. drafted the article with contributions from all the authors.

## Competing interests

The authors declare no competing interests.
