## [Peer Review File · Nature Communications]

REVIEWER COMMENTS

Reviewer #1 (Remarks to the Author):

The authors in this manuscript present the first implementation of a traveling-wave parametric amplifier (TWPA) realized from a Josephson metamaterial of SNAIL elements. The SNAIL element an asymmetric SQUID allows for tuning of the linear and nonlinear properties of the element including the second and third order (Kerr) nonlinearity with an externally applied magnetic field. This element and the tunable nonlinearity provide great flexibility over the typical Josephson junction nonlinearity and has attracted significant attention recently being utilized in novel superconducting qubit designs as well as several proposals for phase matching both three and four-wave mixing processes in TWPAs to realize efficient parametric amplification.

Reported in this manuscript the authors have experimentally demonstrated a TWPA realized from a transmission line composed of 700 SNAIL elements. A property of the SNAIL element is that when flux biased to full frustration the Kerr coefficient can be large and change sign relative to the Kerr coefficient of a Josephson junction, also at the same time the second order nonlinearity goes to zero. This tuning condition allows for efficient phase matching of a four-wave mixing process and a suppression of three wave mixing as well as second harmonic generation of the pump. To suppress the second order nonlinearity due to imperfections in the fabrication, process the authors have physically oriented the SNAIL elements in the TWPA such that magnetic flux biasing is reversed every other SNAIL element to minimize 2nd harmonic generation. Utilizing the mismatch due to chromatic dispersion of the TWPA and the reverse sign of the Kerr coefficient the authors were then able to phase match a four-wave mixing process in the TWPA which resulted in gains of more than 20 dB with a 1dB compression of -98 dBm. This experimental realization of this reverse Kerr approach to phase matching a four-wave mixing process is different from all other experimental approaches presented in literature which utilizes dispersion engineering by introducing resonance elements or photonic bandgaps in the transmission line of the TWPA in which the pump frequency can be tuned close to compensate for the self-phase modulation of the pump. The work presented in this manuscript does not include any such dispersion engineering tactics which has the advantage of a tunable pump as was demonstrated in the manuscript for pumps of 6, 8, 10 GHz, the maximum gain does not occur close to the pump in an unusable area of the bandwidth as with competing designs, there is not a stop band in the bandwidth of the amplifier due to dispersion engineering structures, and ripple in the passband of the amplifier is reduced because there is not a sensitive gain dependence on the spread in parameters of the dispersion engineering resonant structures. The authors have also measured the noise performance of the TWPA to be about twice the standard quantum limit. This noise is in line with conventional TWPA approaches and most likely dependent on the losses in the TWPA presented in the manuscript which are about the same as conventional TWPAs.

Most TWPA realizations in literature today all take the same dispersion engineering approach to phase matching a four-wave mixing process. This manuscript presents a different approach with many advantages utilizing the unique tunable nonlinearity of SNAIL elements which have gained

significant attention recently. I recommend this manuscript for publication in Nature Communications. I do have a couple comments that might strengthen this article.

It would be good to include the wave equation of the transmission line which make up the TWPA. When presenting the dispersion relation eq. 6 the authors indicate that they solved the “transmission” in the absence of nonlinearities. Presentation of the wave equation would make everything including the design of the TWPA clearer.

I also recommend putting in the design parameters of the TWPA such as critical currents of the junctions, r value, and capacitances in the manuscript.

The inductance L of the SNAIL element which is the per-unit length inductance of the transmission line of the TWPA is not defined other than the “flux-tunable inductance”. This L is one of the most important quantities of the circuit. I think defining this would be good.

The gain presented in the manuscript - it is unclear if this is the gain measured as the output signal over the input signal or the increase in the output signal when the pump is turned on and off. It would be good if this can be clarified.

I believe the gain is represented as the increase in output signal when the pump is turned on and off. If this is the case then the gain of this amplifier, if one intends to use it as an amplifier to amplify a weak signal at the input does not reach 20 dB since insertion losses of the TWPA are 6 dB @ 6 GHz and 12 dB at 10 GHz. I would suggest putting the loss data in the manuscript and not in supplementary information since this is of great interest to the reader in determining device performance.

With these losses increasing the length of the TWPA probably will not result in increased gain since the whole TWPA will not be phase matched. Have the authors considered any other approaches to increase the gain to 20 dB gain to compensate a depleting pump.

In this manuscript the authors appreciated the importance of suppressing the second harmonic generation of the pump and improved the device design to suppress this harmonic. How effectively did this approach work? Was the second harmonic measured? Also, the third harmonic, was this signal measured? The generation of the second and third harmonic of the pump might be more detrimental to pump depletion along the TWPA than loss. What is the plasma frequency of this transmission line and how well does that suppress such harmonic generation and higher mixing products? Recent work by Ref 44 and K O’Brien (MIT) have illustrated how it will limit gain and effect noise properties due to higher mixing products which result from these higher harmonics of the pump.

Reviewer #2 (Remarks to the Author):

The authors of the manuscript Kerr reversal in Josephson meta-material and traveling wave parametric amplification present a new way to achieve parametric amplification in a Josephson traveling-wave parametric amplifier (JTWPA) operated in the four-wave mixing (4WM) mode. This new way relies on the possibility to tune the nonlinearity of the line in a regime where its third order (Kerr) term g_4 is negative, and can thus balance the always positive linear dispersion relation that otherwise exists in 4WM-operated JTWPAs. The nonlinearity of the line is implemented with an array of superconducting nonlinear asymmetric inductive elements (SNAILS).

Although the physics of the nonlinearity is not new, the manuscript presents a new device for which the experimental characterization is technically strong, and the theory supporting the data is clear. Overall, the presented amplifier is promising for qubit experiments, and it has better performances than other existing JTWPAs. However, I think that some of the device's performances are a bit underwhelming, while other engineering solutions are not advertised enough.

There are three big advantages to their new JTWPA design and operation: (i) the transmission line does not need a phase matching structure (resonant or periodic loading) that is otherwise implemented in any other TWPAs. This is a technical simplification that avoids impedance mismatch issues along the line, and therefore minimizes the generation of gain ripples in the gain profile. I note however that beyond 8 GHz, their gain profile does contain non-negligible ripples. Could the authors comment on their origin and how to further damp them? (ii) the amplification bandwidth is tunable in situ, by just changing the rf pump frequency. Although their devices already present a wide (~ 4 GHz) amplification bandwidth, they can adjust the center of this band, from a demonstrated 6 GHz to 10 GHz. This is practically useful for qubits experiments. (iii) They implemented a gradiometric-type of structure in the JTWPA, where the orientation of the SNAIL loops alternates. The device is therefore intrinsically protected against stray magnetic field, and they don't need to shield it. In addition, at the flux where the JTWPA is operated, g_4 is first order insensitive to flux variations, therefore so is the phase matching condition. This, in my opinion, should be advertised a bit more.

In principle, their implementation has another advantage: they can operate the nonlinearity at minimum g_4 , while canceling g_3 (the second order nonlinearity), which avoids the parametric amplification to be contaminated with 3WM processes, and therefore increases the power handling of the device. But in that regard, their demonstration is underwhelming, because they only demonstrate -98 dBm of (input referred) saturation power, which is on the low end of what parametric amplifiers leveraging the SNAIL can do. Is it because of the low critical current of the junctions? For an application where the saturation power is the main concern, wouldn't I be better off with operating the same device in 3WM, leveraging g_3 and canceling g_4 ?

Also, they advertise the absence of gap in the amplification bandwidth, which exists in other 4WM JTWPAs. This is true, but if one wants to avoid having a gap within the amplification band, one could simply switch to a 3WM operating mode.

Shall these few comments be addressed by the authors, along with the more detailed questions I have below, I think this manuscript would be fit for Nature Communications.

MANUSCRIPT

% DEVICE DESCRIPTION AND OPERATION

- Why is Fig.1 presented before Fig.2? The authors first discuss Fig.2 in the article.
- The authors should specify the physical length of the device
- g_3 & g_4 are defined as 'coefficients', but I think they can be better characterized: they are the rates at which the 3WM and 4WM interactions happen, respectively.
- The sentence "Since g_3 is an odd... discussion in this article" is unclear; please reformulate.
- First column, bottom of page 3: Fig.2(d) should be Fig.2(a)
- below Eq.3: the subscripts s & i should not be capitalized
- The authors should specify (in appendix or supplementary) the full expression of Δk_{Kerr} .
- The authors should define $\omega_0 = 1/\sqrt{LCg}$ in the main text.
- second column, page 4: Fig.2(a) should be Fig.2(c).
- First column, page 5, last paragraph: is the coil around or below the chip ?

% Fig 1

- panel (a): you could underline the fact that the red arrows indicate how strong is the self-phase modulation (SPM) for the pump, and the cross-phase modulation (XPM) for the signal and idler. It is interesting to notice that the SPM strength is a bit weaker than the XPM, which allows the three dots to then be aligned.

% Fig 2

- panel (a): it would be interesting to also plot g_3 .
- in the caption, can you specify the values of r & I_0 ?

% Fig 3

- What do I do if I want to have a flatter gain profile (instead of one that has an M shape)?
- Can you comment on the ripples observed in particular at high frequency?
- The higher frequency range of the M shape consistently presents a higher gain, why is that?

% NOISE CALIBRATION

- the subscripts in Eq. 7 could be simplified
- the last paragraph before the conclusion could be put in an appendix or in the supplementary, along with Fig.4 (c) and (d). Although important, it is not part of the main message of the manuscript.

SUPPLEMENTARY

% Fig.2

- In the inset, there a change in transmission between the zero flux operation and the half flux operation, why?

- What was the nominal choice for r ? Was it $r=0.1$?

The authors in this manuscript present the first implementation of a traveling-wave parametric amplifier (TWPA) realized from a Josephson metamaterial of SNAIL elements. The SNAIL element an asymmetric SQUID allows for tuning of the linear and nonlinear properties of the element including the second and third order (Kerr) nonlinearity with an externally applied magnetic field. This element and the tunable nonlinearity provide great flexibility over the typical Josephson junction nonlinearity and has attracted significant attention recently being utilized in novel superconducting qubit designs as well as several proposals for phase matching both three and four-wave mixing processes in TWPAs to realize efficient parametric amplification.

Reported in this manuscript the authors have experimentally demonstrated a TWPA realized from a transmission line composed of 700 SNAIL elements. A property of the SNAIL element is that when flux biased to full frustration the Kerr coefficient can be large and change sign relative to the Kerr coefficient of a Josephson junction, also at the same time the second order nonlinearity goes to zero. This tuning condition allows for efficient phase matching of a four-wave mixing process and a suppression of three wave mixing as well as second harmonic generation of the pump. To suppress the second order nonlinearity due to imperfections in the fabrication, process the authors have physically oriented the SNAIL elements in the TWPA such that magnetic flux biasing is reversed every other SNAIL element to minimize 2nd harmonic generation. Utilizing the mismatch due to chromatic dispersion of the TWPA and the reverse sign of the Kerr coefficient the authors were then able to phase match a four-wave mixing process in the TWPA which resulted in gains of more than 20 dB with a 1dB compression of -98 dBm. This experimental realization of this reverse Kerr approach to phase matching a four-wave mixing process is different from all other experimental approaches presented in literature which utilizes dispersion engineering by introducing resonance elements or photonic bandgaps in the transmission line of the TWPA in which the pump frequency can be tuned close to compensate for the self-phase modulation of the pump. The work presented in this manuscript does not include any such dispersion engineering tactics which has the advantage of a tunable pump as was demonstrated in the manuscript for pumps of 6, 8, 10 GHz, the maximum gain does not occur close to the pump in an unusable area of the bandwidth as with competing designs, there is not a stop band in the bandwidth of the amplifier due to dispersion engineering structures, and ripple in the passband of the amplifier is reduced because there is not a sensitive gain dependence on the spread in parameters of the dispersion engineering resonant structures. The authors have also measured the noise performance of the TWPA to be about twice the standard quantum limit. This noise is in line with conventional TWPA approaches and most likely dependent on the losses in the TWPA presented in the manuscript which are about the same as conventional TWPAs.

Most TWPA realizations in literature today all take the same dispersion engineering approach to phase matching a four-wave mixing process. This manuscript presents a different approach with many advantages utilizing the unique tunable nonlinearity of SNAIL elements which have gained significant attention recently. I recommend this manuscript for publication in Nature Communications. I do have a couple comments that might strengthen this article.

We thank referee 1 for the constructive and valuable comments and for pointing out the scientific importance of our demonstration of traveling wave parametric amplification with reversed Kerr phase matching mechanism, and recommending it for publication.

In the following, we address the comments and questions raised by the referee point-by-point.

1. It would be good to include the wave equation of the transmission line which make up the TWPA. When presenting the dispersion relation eq. 6 the authors indicate that they solved the “transmission” in the absence of nonlinearities. Presentation of the wave equation would make everything including the design of the TWPA clearer. I also recommend putting in the design parameters of the TWPA such as critical currents of the junctions, r value, and capacitances in the manuscript.

We thank the referee for this comment, we added the semi classical wave equation (eq 7), which is used to obtain the dispersion relation in the main text of the article. We have described the linear characterization along with circuit parameters in the supplementary information, we agree with the referee that it would be useful to have them in the manuscript, we have added these details in methods along with device unit cell description.

2. The inductance L of the SNAIL element which is the per-unit length inductance of the transmission line of the TWPA is not defined other than the “flux-tunable inductance”. This L is one of the most important quantities of the circuit. I think defining this would be good.

We have clarified the definition of L in the main text and added explicit expression in methods. A detailed flux dependent characterization of L is reported in supplementary information.

3. The gain presented in the manuscript - it is unclear if this is the gain measured as the output signal over the input signal or the increase in the output signal when the pump is turned on and off. It would be good if this can be clarified. I believe the gain is represented as the increase in output signal when the pump is turned on and off. If this is the case then the gain of this amplifier, if one intends to use it as an amplifier to amplify a weak signal at the input does not reach 20 dB since insertion losses of the TWPA are 6 dB @ 6 GHz and 12 dB at 10 GHz. I would suggest putting the loss data in the manuscript and not in supplementary information since this is of great interest to the reader in determining device performance.

We thank the referee for pointing out the need for clarification here, as suggested we have specified the definition of the TWPA gain as the difference in transmission when pump is on and when pump is off. We also added the plot of the losses in the manuscript. We would also like to point out that the noise temperature is a robust qualifier of performance of the amplification process, which includes contribution of losses to the device performance.

4. With these losses increasing the length of the TWPA probably will not result in increased gain since the whole TWPA will not be phase matched. Have the authors considered any other approaches to increase the gain to 20 dB gain to compensate a depleting pump.

The referee accurately observes the limitation posed by losses to TWPA performance. Ideally, improvement can be achieved by reduction of losses by using a dielectric with lower loss tangent, for example SiN_x or a:SiH (ref 51, manuscript). Other promising approach would involve increasing power handling capacity of the device by increasing critical current of the Josephson junctions. Both these approaches require tackling significant fabrication challenges, and are beyond the scope of this article.

5. In this manuscript the authors appreciated the importance of suppressing the second harmonic generation of the pump and improved the device design to suppress this harmonic. How effectively did this approach work? Was the second harmonic measured? Also, the third harmonic, was this signal measured? The generation of the second and third harmonic of the pump might be more detrimental to pump depletion along the TWPA than loss. What is the plasma frequency of this transmission line and how well does that suppress such harmonic generation and higher mixing products? Recent work by Ref 44 and K O’Brien (MIT) have illustrated how it will limit gain and effect noise properties due to higher mixing products which result from these higher harmonics of the pump.

The referee raises an interesting question regarding contribution of different mechanisms to the degradation of amplification performance of TWPA. As depicted in Fig 4 of the main text, noise added by the TWPA during amplification is in very good agreement with simulations performed assuming dielectric losses as the only source of degradation. This is a strong indication that dielectric losses are primary limitation to improvement of amplification performance in the presented device, and also that higher harmonics are adequately suppressed. In a device with lower losses, higher order processes would indeed become important, requiring strong suppression to reach quantum limited noise performance as reported in the references mentioned by the referee.

The plasma frequency of the presented device at optimal flux point is ~ 30 GHz, which offers natural suppression of higher harmonic generation due to phase mismatch. Quantitatively, the phase mismatch offered by this curvature for second harmonic generation $\Delta k_{2H} := k_{2H} - 2k_p \sim 20\Delta k_{amp}$ and for third harmonic generation $\Delta k_{3H} := k_{3H} - 3k_p \sim 100\Delta k_{amp}$, where Δk_{amp} is the phase mismatch offered by dispersion curvature for four wave mixing amplification process.

We performed WRSPICE simulations to better understand the suppression of second harmonic generation by design (alternating geometry/flux parity), as depicted in Fig. 3 of supplementary material. A quantitative experimental study of the same would involve fabrication and measurement of devices with and without the alternating geometry inversion. As a platform for novel nonlinear optics experiments, it can be interesting to understand second and third harmonic generation in such Josephson meta-materials, however, such study is beyond the scope of this article.

The authors of the manuscript Kerr reversal in Josephson meta-material and traveling wave parametric amplification present a new way to achieve parametric amplification in a Josephson traveling-wave parametric amplifier (JTWPA) operated in the four-wave mixing (4WM) mode. This new way relies on the possibility to tune the nonlinearity of the line in a regime where its third order (Kerr) term g_4 is negative, and can thus balance the always positive linear dispersion relation that otherwise exists in 4WM-operated JTWPAs. The nonlinearity of the line is implemented with an array of superconducting nonlinear asymmetric inductive elements (SNAILs). Although the physics of the nonlinearity is not new, the manuscript presents a new device for which the experimental characterization is technically strong, and the theory supporting the data is clear. Overall, the presented amplifier is promising for qubit experiments, and it has better performances than other existing JTWPAs. However, I think that some of the device's performances are a bit underwhelming, while other engineering solutions are not advertised enough.

We thank referee 2 for the constructive and valuable comments and for recognizing the solidity of the experimental results and of the theory supporting them. In the following, we address the comments and questions raised by the referee point-by-point.

1. There are three big advantages to their new JTWPA design and operation: (i) the transmission line does not need a phase matching structure (resonant or periodic loading) that is otherwise implemented in any other TWPAs. This is a technical simplification that avoids impedance mismatch issues along the line, and therefore minimizes the generation of gain ripples in the gain profile. I note however that beyond 8 GHz, their gain profile does contain non-negligible ripples. Could the authors comment on their origin and how to further damp them? (ii) the amplification bandwidth is tunable in situ, by just changing the rf pump frequency. Although their devices already present a wide (4 GHz) amplification bandwidth, they can adjust the center of this band, from a demonstrated 6 GHz to 10 GHz. This is practically useful for qubits experiments. (iii) They implemented a gradiometric-type of structure in the JTWPA, where the orientation of the SNAIL loops alternates. The device is therefore intrinsically protected against stray magnetic field, and they don't need to shield it. In addition, at the flux where the JTWPA is operated, g_4 is first order insensitive to flux variations, therefore so is the phase matching condition. This, in my opinion, should be advertised a bit more.

We thank the referee for pointing out the above three advantage of our Reversed Kerr TWPA.

(i) As noted by the referee absence of impedance mismatch arising from dispersion engineering leads to lower gain ripples in the amplification profiles. The gain ripples at high frequencies are primarily arising from the sub-optimal performance of the packaging (sample holder) of the device at high frequencies. Improving performance of the sample boxes at high frequencies is an engineering challenge we are currently working on.

(ii) We agree with the referee that this ability of dynamic tuning of amplification band by changing pump frequency makes this amplifier very useful for many experiments, including qubit measurements, and believe is one of the key advantages of reversed Kerr implementation over traditional TWPAs.

(iii) We thank the referee for this very constructive comment, we have stressed this aspect more in the main text of the manuscript.

2. In principle, their implementation has another advantage: they can operate the nonlinearity at minimum g_4 , while canceling g_3 (the second order nonlinearity), which avoids the parametric amplification to be contaminated with 3WM processes, and therefore increases the power handling of the device. But in that regard, their demonstration is underwhelming, because they only demonstrate -98 dBm of (input referred) saturation power, which is on the low end of what parametric amplifiers leveraging the SNAIL can do. Is it because of the low critical current of the junctions? For an application where the saturation power is the main concern, wouldn't I be better off with operating the same device in 3WM, leveraging g_3 and canceling g_4 ?

As identified by the referee, in the device presented, amplifier saturation is mainly limited by our fabrication capabilities (critical current); which is comparable to state of the art in Josephson meta-material based parametric amplifiers. The four wave mixing process utilized for amplification is very efficient in power conversion and the upper limit on saturation power of Josephson meta-material based amplifiers is dictated by the critical current of Josephson junctions, which limits the maximum input pump power. For an application where the saturation power is the primary concern, TWPAs implemented utilizing kinetic inductance can be very promising. However, they require very high pump powers (typically 4-5 orders of magnitude higher than JTWPAs) which might add requirement of pump cancellation schemes to avoid saturation of the next stages of amplification.

We agree with the referee that one of the very promising perspectives of SNAIL-based meta-materials is 3WM parametric amplification. We are actively working in this direction, however, in the device presented in the manuscript, we deliberately suppressed the 3WM contribution by designing the SNAIL elements with alternating magnetic flux polarity. In this way, the magnetic flux is inverted every other SNAIL element suppressing the 2nd order non-linearity, thus spurious 3WM processes. A similar device, but without the alternating inversion of the SNAIL polarity would exhibit full range of both 3WM and

4WM non-linearities. However, it should be noted that realization of 3WM gain requires adoption of strategies to mitigate unwanted 3WM processes i.e. second harmonic generation and frequency up-conversion, which are the dominant processes and can deplete pump and degrade signal. These mitigation strategies usually involve dispersion engineering with gaps, with fixed pump frequency and amplification band.

3. Also, they advertise the absence of gap in the amplification bandwidth, which exists in other 4WM JTWPAs. This is true, but if one wants to avoid having a gap within the amplification band, one could simply switch to a 3WM operating mode.

We agree with the referee that in principle a gap in amplification band can be indeed avoided by utilizing 3WM process for amplification. However, we would like to point out that in 3WM TWPAs, for example, the one reported in ref [31], dispersion engineering near pump frequencies is still necessary to suppress spurious 3WM processes. This gap engineering fixes the optimal pump frequency and amplification band by design.

4. Why is Fig.1 presented before Fig.2? The authors first discuss Fig.2 in the article.

We thank the referee for pointing out this discrepancy. We arranged the manuscript to present the concept of reversed Kerr phase matching (Fig 1) first and then its implementation (Fig 2). We have corrected the order in which these figures are referenced in the article.

5. The authors should specify the physical length of the device.

The physical length of the JJ meta-material is 6mm, we added the same in main text of the article.

6. g_3 & g_4 are defined as ‘coefficients’, but I think they can be better characterized: they are the rates at which the 3WM and 4WM interactions happen, respectively.

We agree with the referee and amended the definition of the same.

7. The sentence “Since g_3 is an odd... discussion in this article” is unclear; please reformulate.

We have changed the wording of the explanation to make it clearer.

8. First column, bottom of page 3: Fig.2(d) should be Fig.2(a)

We have made the necessary correction.

9. below Eq.3: the subscripts s & i should not be capitalized

We have made the necessary correction.

10. The authors should specify (in appendix or supplementary) the full expression of $\Delta_k \text{Kerr}$.

We have added the exact expression for Δk_{Kerr} in the manuscript.

11. The authors should define $\omega_0 = 1/\sqrt{LCg}$ in the main text.

We have implemented the suggestion.

12. second column, page 4: Fig.2(a) should be Fig.2(c).

We have made the necessary correction.

13. First column, page 5, last paragraph: is the coil around or below the chip ?

The coil is mounted on top of the sample holder, we have mention the same in the paragraph.

14. Fig 1

- panel (a): you could underline the fact that the red arrows indicate how strong is the self-phase modulation (SPM) for the pump, and the cross-phase modulation (XPM) for the signal and idler. It is interesting to notice that the SPM strength is a bit weaker than the XPM, which allows the three dots to then be aligned.

We have implemented the suggestion.

15. Fig 2

- panel (a): it would be interesting to also plot g_3 .
- in the caption, can you specify the values of r & I_0 ?

- In the panel (a), we made a conscious choice to not include the plot of g_3 , as the device presented in manuscript has alternating SNAIL geometry (flux polarity), and in principle should not exhibit any g_3 .

- We have added the values of r & I_0 in the caption.

16. Fig 3

- What do I do if I want to have a flatter gain profile (instead of one that has an M shape)?
- Can you comment on the ripples observed in particular at high frequency?
- The higher frequency range of the M shape consistently presents a higher gain, why is that?

- This is a very interesting inquiry. The "M" shape of the gain comes from the fact that the phase matching is optimal at two points on either side of the pump, as depicted in Fig. 1 of main text. This profile can be modified by changing the shape of dispersion, i.e. by operating with stronger curvature the optimally phase matched frequencies can be brought closer. However, it should be noted that the "M" shape is characteristic of reversed Kerr phase matching technique, and can not be completely eliminated.

- The gain ripples at high frequencies are mainly arising from the sub-optimal performance of the packaging (sample box) of the device at high frequencies.

- This is indeed an interesting feature of the gain profile. The gain obtained at pairs of frequencies equidistant from the pump, in an ideal lossless reversed Kerr TWPA would be identical. However, practically, asymmetric loss experienced by waves traveling at different frequencies breaks this gain symmetry. This dynamics is well captured by the distributed model presented in the manuscript, used to obtain theory curves in Fig. 3 (a) of main text.

17. NOISE CALIBRATION

- the subscripts in Eq. 7 could be simplified
- the last paragraph before the conclusion could be put in an appendix or in the supplementary, along with Fig.4 (c) and (d). Although important, it is not part of the main message of the manuscript.

-We agree with the referee that the subscripts look long in at first look, however, we expressed it in this way to make it easily understandable and to indicate that its valid not only for thermal source, but for any broadband noise source, for example a shot noise tunnel junction.

We thank the referee for this suggestion. We understand that this is not the main message of the manuscript, however we had made a conscious choice to highlight this common mistake in measuring of noise performance as it can help in making comparisons with performance of other amplifiers.

18. SUPPLEMENTARY - Fig.2

- In the inset, there a change in transmission between the zero flux operation and the half flux

operation, why?

- What was the nominal choice for r ? Was it $r=0.1$?

- The change in flux changes the inductance, and hence dispersion through the meta-material. This can also be seen as increase in effective length of the lossy medium, which leads to lower transmission.

- The design value of r for the presented device was 0.1.

REVIEWERS' COMMENTS

Reviewer #1 (Remarks to the Author):

The authors have addressed my concerns in their manuscript. I would of liked to see more in regards to suppression of higher harmonics which the authors have attempted to do, and mentioned in the manuscript, however did not present any data showing the outcome of these efforts.

With the noise measurements the authors only supply noise from the noise source below the noise temperature of the TWPA. How come larger noise temperature from the noise source were not supplied to the input of the TWPA? With a saturation power of ~ -98 dBm did the amplifier saturate on such noise? Can a reason for the limited range in noise source temperature for measurement be discussed in the manuscript or supplementary material?

I recommend publication of this manuscript.

Reviewer #2 (Remarks to the Author):

The authors have addressed all the questions and concerns I had in the previous round of feedback. I commend their efforts, and am happy to see the manuscript published in Nature Communications.

Referee response to author replies

The authors have addressed my concerns in their manuscript. I would of liked to see more in regards to suppression of higher harmonics which the authors have attempted to do, and mentioned in the manuscript, however did not present any data showing the outcome of these efforts.

With the noise measurements the authors only supply noise from the noise source below the noise temperature of the TWPA. How come larger noise temperature from the noise source were not supplied to the input of the TWPA? With a saturation power of -98 dBm did the amplifier saturate on such noise? Can a reason for the limited range in noise source temperature for measurement be discussed in the manuscript or supplementary material?

I recommend publication of this manuscript.

We thank referee 1 for the insightful discussion and for recommending the article for publication.

We agree with the referee that as a platform for novel nonlinear optics experiments, it can be interesting to understand higher harmonic suppression in such Josephson metamaterials. However, a detailed study would require fabrication and measurement of devices without implementation of harmonic suppression, and is beyond the scope of this article.

The referee has correctly identified the limitation of using broadband noise source for noise calibration of broadband amplifiers. Although the experimental setup that we had designed had the capacity to heat up the noise source up to 1 K, we had to limit the noise measurements to few 100 mK. Due to the broadband nature of the amplifier, the noise source heated above 0.5 K starts to produce saturation effects, with total power emitted in the active band of amplifier reaching -98 dBm.

We thank the referee again for pointing out that these details can be interesting for the readers; we have added these details in the section discussing noise measurements in the article.